# Obscurant Segmentation in Long Wave Infrared Images Using GLCM Textures

**DOI:** 10.3390/jimaging8100266

**Published:** 2022-09-30

**Authors:** Mohammed Abuhussein, Aaron Robinson

**Affiliations:** Electrical and Computer Engineering, University of Memphis, Memphis, TN 38152, USA

**Keywords:** segmentation, thermal, infrared, LWIR, unsupervised, texture, occlusion, obscurants, clouds, GLCM, Gabor, MRF

## Abstract

The benefits of autonomous image segmentation are readily apparent in many applications and garners interest from stakeholders in many fields. The wide range of benefits encompass applications ranging from medical diagnosis, where the shape of the grouped pixels increases diagnosis accuracy, to autonomous vehicles where the grouping of pixels defines roadways, traffic signs, other vehicles, etc. It even proves beneficial in many phases of machine learning, where the resulting segmentation can be used as inputs to the network or as labels for training. The majority of the available image segmentation algorithmic development and results focus on visible image modalities. Therefore, in this treatment, the authors present the results of a study designed to identify and improve current semantic methods for infrared scene segmentation. Specifically, the goal is to propose a novel approach to provide tile-based segmentation of occlusion clouds in Long Wave Infrared images. This work complements the collection of well-known semantic segmentation algorithms applicable to thermal images but requires a vast dataset to provide accurate performance. We document performance in applications where the distinction between dust cloud tiles and clear tiles enables conditional processing. Therefore, the authors propose a Gray Level Co-Occurrence Matrix (GLCM) based method for infrared image segmentation. The main idea of our approach is that GLCM features are extracted from local tiles in the image and used to train a binary classifier to provide indication of tile occlusions. Our method introduces a new texture analysis scheme that is more suitable for image segmentation than the solitary Gabor segmentation or Markov Random Field (MRF) scheme. Our experimental results show that our algorithm performs well in terms of accuracy and a better inter-region homogeneity than the pixel-based infrared image segmentation algorithms.

## 1. Introduction

Segmentation is a widely studied field in computer vision. Over the past decade, significant advances have been introduced to address the application of semantic segmentation to vehicle-based imagery. This is especially relevant in applications involving autonomous vehicles as increased segmentation proficiency directly addresses many of the challenges facing self driving cars and autonomous vehicles in general. These challenges include highly variable road conditions, fluctuating weather patterns, and unpredictable traffic scenarios. All three cases pose challenges to autonomous vehicle function due to their resulting effects on traffic signal, road boundary and lane markings visibility. Weather conditions degrade feature collection and mapping capabilities in degraded visual environments and random traffic environments add unspecified shapes, speeds, and unpredictable movement to autonomous systems attempting to complete complicated segmentation and classification processes [1,2]. As the fields of machine and deep learning gain large popularity, more models and solutions based on those methods have been introduced to provide enhanced results supporting identification of objects in scenery and autonomous vehicle function. In 2015, J.Long et al. introduced the first Fully Convolutional Network (FCN) to perform semantic segmentation on colored images [3]. This model reached 62.2% mean Intersection over Union (mIoU) accuracy on the 2012 PASCAL dataset. Subsequent studies have introduced improvements on FCN in [4] with 69.8% mIoU. Additional models have been introduced since in [5,6,7,8,9,10,11,12,13,14,15] which can produce significantly improved results. All of the previously mentioned models have been trained and tested on true color images (RGB images) in large datasets such as, VOC and PASCAL. The performance of Convolutional Neural Networks (CNNs) can indeed be accurate on trained environments for a familiar image type. However, CNNs typically fail to provide accurate labels when dealing with new type of imagery or scenes, e.g., thermal images, or scenes in rural roads. Thermal imagery provides immense assistance in seeing in dark environments since it depends on heat emitted from objects rather than light reflection. In addition to the poor performance when testing on new data, CNNs are typically prone to prediction accuracy reductions in low contrast images [16]. The resulting low accuracy causes unpredictable results in applications involving dispersing clouds or hazy images. Finding the appropriate segmentation algorithm to separate occlusion and haze from the rest of the frame was the motivation to start the work. Supporting the necessary autonomous vehicle functions of road, sky, and cloud classification in addition to the inclusion of appropriate measures to ensure vehicle operation in inclement environments such as smoke or fog provided additional incentives to complete the effort. The most evident advantage being that identification of occluded areas beyond improvement implies that standard image processing techniques can then be applied to the rest of the frame to ensure maximum benefit/processing ratios. We start the work by proposing an approach to provide binary mask classification for obscurant clouds in thermal images. In this paper, we introduce a hard-labeled method to provide semantic segmentation for vehicle-based thermal imagery. The contents of development is summarized as follows: The “Related Works” section will provide background/summaries of some of the most widely known and implemented algorithms. The subsequent sections address the application scenario, the developed algorithmic enhancements, and the results of its implementation, respectively. The paper then finishes with some applicable conclusions and ideas for possible future research directions.

## 2. Related Work

Geographic information systems (GIS) and satellite imagery play crucial roles for automotive cars function through updated and advanced imagery systems. Fully automated algorithms characterized by high accuracy and fast responding systems can save time and effort of updating the spatial data in the maps and applications. Research results are dependent upon enhancement operations performance along with the inclusion of various types of images. Moreover, parameter calculations for homogeneity and local segmentation based on the template are also relevant [17].

Textures, road colours, and lighting are important factors for road detection, and image segmentation in Automatic Driver Assistant Systems (ADAS). Researchers used road segmentation and dominant edge detection approaches [18]. When trying to apply road detection algorithms on Long Wave Infrared (LWIR) images, we noticed a reduction in the performance due to varying gamma values, shading, and obscurants. Several treatments have addressed the shading and gamma values by performing gamma-matching and image normalization. In this paper, we aim to solve the problem of segmenting the obscurants as a first step to providing better segmentation results in the rest of the image. This provides the ability to locally process the obscurant cloud without distorting the other not obscured pixels.

From previous research, we noticed that most unsupervised algorithms that rely on pixel values or edges do not perform well when dealing with clouds of obscurants. In order to build a segmentation algorithm for obscurant clouds, we had to look for, study, and evaluate cloud and texture segmentation algorithms.

### 2.1. Cloud Segmentation

After inspecting the occlusion faced in this problem, we found that the occlusion resembles a cloud in most cases. Therefore, we resorted to reviewing sky/cloud segmentation algorithms. In this regard, various techniques have been introduced to segment cloud pixels. The cloud detecting algorithms can exhibit varying degrees of performance due to environmental conditions. Therefore, diversified approaches to better cope with the issues underlying the image segmentation are required. Image-based cloud segmentation is a big challenge in this regard. For example, techniques remote sensing techniques of the earth’s atmosphere are also remarkable. The distant clouds can be processed through the practical approach of semantic segmentation. This task becomes challenging, i.e., specifically for the clouds and the weather forecasts, as the clouds always have asymmetrical shapes and structures. Several computer vision approaches have demonstrated excellent performance in tackling the segmentation problem.

Tulpan et al., 2017 introduced a framework to classify cloud pixels by extracting six image moments from a window of size *W*, where the center is the pixel to be labeled. The extracted moments with a window centered at the pixel (i,j) resembles a convolution operation where it can be used to extract the area of the image, edges (both vertical and horizontal), diagonal edges, and the elongation in both *x*, and *y* directions. These features are then used to train a classifier to predict cloud pixels. This approach was developed and tested in visible light spectrum, and used the Red/Blue channels information to test the efficacy of the classifier [19].

Dev et al., 2017 presented a color selection scheme to define the predominant feature for sky/cloud segmentation in [20]. Many techniques have been used for the sky and cloud segmentation. RGB, HSV, YIQ, CIE, three forms of red-blue combinations (R/B, R−B, B−RB+R), and chroma C=max(R, G, B)−min(R, G, B). A color model can help to facilitate the task of inter-pixel differentiation. For clustering the cloud images, a fuzzy algorithm has been used. For verification, the efficacy of specific color channels in detecting cloud pixels, 1D and 2D clustering results are presented across all 16 color channels for all the images of HYTA and evaluated using precision, recall, and F-score. In [21,22,23], the authors propose a deep learning based approaches to segment sky/cloud images. The model proposed in 2019 in [21] achieved a binary classification accuracy of 89%, the proposed approach from 2019 in [22] achieved a 90% accuracy, and [23] published in 2022 achieved 95% precision. However, similar to the previously listed approaches, the network architecture proposed is designed using visible light images (RGB channels). In 2020 An elaborate study was done on several algorithms for cloud segmentation [24]. They consider the multi-color criterion method, the region growing approach, the Hybrid Threshold Algorithm HYTA, the improved HYTA or HYTA+, a Clear-Sky-Library (CSL) approach, and a Fully Connected Network (FCN). This study reported accuracy higher than 90% for CSL, HYTA+ and FCN. The FCN algorithm tested in [24] was based on the model introduced by The Visual Geometry Group (VGG) in [25], and was pre-trained on ImageNet Large Scale Visual Recognition Competition (ILSVRC) [26] on RGB images. The network was fine-tuned using a binary segmentation masks (cloud or clear) in RGB as well. Thus, far, all the previously investigated, although robust in their domain, failed in our application.

### 2.2. Texture Analysis

Segmentation of textures necessitates the choice of good texture-specific features with excellent discriminating power. In general, techniques for extracting texture features are categorized into three main classifications: spectral, structural, and statistical. In spectral techniques, the textured image, as Madasu and Yarlagadda pointed out in [27], is changed into the frequency domain. After that, extracting the texture features can be carried out by assessing the power spectrum. In structural–based feature extraction techniques, the fundamental facet of texture, known as texture primitive, is utilized in forming more intricate patterns of texture through the application of rules that stipulate how texture patterns are generated. Lastly, in statistical techniques, texture statistics, for instance, the moments of the gray-level histogram, are founded upon gray-level co-occurrence matrix and are calculated for discriminating different textures [27]. Over the years, many different methods have been developed for texture-based segmentation. The main ones include Gabor filters, MRF, and Gray Level Co-Occurrence Matrix methods.

Markov Random Field (MRF) is a highly sophisticated texture-based segmentation method. Regions in natural images are usually homogeneous. Pixel homogeneity means that adjacent pixels often have similar properties. For instance, these properties include common characteristics such as texture, color, and intensity. MRF captures such contextual constraints. MRF is extensively studied and also has a solid theoretical background. According to [28], the MRF segmentation can only be applied to a Markovian image. A Markovian image is an image where the probability distribution of gray levels depends on the neighboring pixels’ gray levels, represented by Gibbs fields.

A Gaussian kernel function modulated by a sinusoidal plane wave is a two-dimensional (2D) Gabor filter in the spatial domain. It is notable that frequency and orientation representations of Gabor filters are comparable to those of the human visual system and are suitable for texture discrimination and representation [29]. Gabor response segmentation assumes that all pixels within the same textured region will provide similar decomposition to different Gabor kernels. All pixels within a region of the same texture are aggregated together.

Both MRF and Wavelets perform well when looking for periodic patterns. This characteristic is beneficial in distinguishing one texture from another. However, for the task of segmenting obscurants in thermal images, and due to the highly correlated pixel values and the resemblance in the different regions and textures, algorithms that assume similarity in the region as a basis for segmentation return inaccurate results.

Grey Level Co-Occurrence Matrix (GLCM) is the indexing of how often a combination of pixel values occur in a certain order in am image. At first glance, GLCM sounds like the spatial correlation between pixels. However, GLCM provides a deeper look into the similarity, dissimilarity, homogeneity, the energy, and the entropy. All the locally extracted features are calculated in four directions at a set distance from the reference pixel. These features will be used to train a classifier to predict if a pixel is a part of an obscurant or a part of the local object or background. In the next section, we will delve into more details about each feature calculated by the GLCM and the classifier model.

## 3. GLCM Features

Haralick first introduced GLCM as a set of features for image classification in [30]. The proposed set of features contained texture properties about the image. These characteristic features where extracted from local blocks surrounding the pixels being analyzed. The Features are calculated based on the Co-Occurrence matrix *P*. The co-occurrence matrix contains information about combination of neighboring pixels withing the local block at a distance *d*, and an orientation θ. Features such as homogeneity similarity, dissimilarity co-occurrence, entropy, and energy are then calculated for the local block using elements P(i,j) from the matrix *P*. The GLCM features are second order features, meaning they describe the relationship between pairs of pixels. Each calculation is done for pairs of pixels, the reference and the neighboring pixel. The level of co-occurrence describes how many times an occurrence happens between the reference and the neighboring pixel, and will result in N×N square, and symmetrical matrix where N is the width of the histogram, or the maximum value in the image. This will result in rather large arrays for 16-bit images of size 65,536 × 65,536. Therefore, the image resolution will need to be reduced to reduced the size of the co-occurrence matrix. Typically, the image’s bit-depth is reduced to 4-bits where the resultant matrix is the size of 16×16. The final GLCM matrix is expressed as a set of probabilities by normalizing each element in the matrix. The normalization process is done by dividing each element by the sum of all elements. Most of the texture quantities such as the entropy and the dissimilarity are weighted average of the normalized GLCM matrix. The co-occurrence probability is expressed as:(1)$$p\left(i,j\right)=\frac{P(i,j)}{\sum_{i}\sum_{j}P(i,j)}$$
Next we will describe each individual feature. To simplify the explanation and how each feature plays a role in texture classification, we grouped the features based on the way the features are calculated into thee into three groups, the pixel intensity features, the regularity feature, and the statistical descriptors.

### 3.1. Regularity Features Group

This set of features is related to the order/transitions in pixel values in a specific window.

#### 3.1.1. Angular Second Moment

Is a measure of homogeneity of the image. An image with very few transitions is considered homogeneous and the ASM value is expected to be large compared to an image with more transitions. The ASM is calculated as the following:(2)$$ASM=\sum_{i}\sum_{j}{\left(\frac{p(i,j)}{R}\right)}^{2}$$

#### 3.1.2. Entropy

The entropy describes the amount of information is withing the image block being analyzed. The entropy *E* is calculated as the following:(3)$$Entropy=-\sum_{i}\sum_{j}p(i,j)log(p(i,j))$$

### 3.2. Pixel Intensity Group

This group describes features based on the distance from the diagonal of the *P* matrix.

#### 3.2.1. Contrast

The contrast is the distance from the diagonal in *P*. Which emphasizes the number of neighboring pixels that have the same value and it calculated as the following:(4)$$Contrast=\sum_{i,j}{\left(i-j\right)}^{2}P\left(j,j\right)$$

#### 3.2.2. Dissimilarity

Similar to the contrast, the dissimilarity is a measure of difference between the weights for the difference in the neighboring pixels:(5)$$Dissimilarity=\sum_{i,j}\left|i-j|P(j,j\right)$$

#### 3.2.3. Homogeneity

Different from the contrast and the dissimilarity, the homogeneity has values between 0 and 1 and it puts more weight to the neighboring pixels with similar values:(6)$$Homogeneity=\sum_{i,j}\frac{p(i,j)}{1+{\left(i-j\right)}^{2}}$$

### 3.3. Statistical Description of GLCM

Similar to the mean, standard deviation and the correlation in common statistics. However, these are calculated on the *P* matrix values instead on the image intensities.

#### 3.3.1. GLCM Mean

This mean represents the average frequency of a certain neighboring pixel combination weighed by the pixel intensity such that:(7)$$\mu_{i}=\sum_{i,j}iP(i,j)$$

#### 3.3.2. Correlation

The correlation is a representation of how much each pixel value is dependent on the neighboring values:(8)$$Corr=\sum_{k=2\times N}\frac{p(i,k)(p(j,k))}{p_{x}\left(i\right)p_{y}\left(j\right)}$$
where $p_{x}\left(i\right),p_{y}\left(j\right)$ is the sum of the GLCM values along the rows, columns, respectively. Such that: $p_{x}\left(i\right)=\sum_{j}p\left(i,j\right)$ and $p_{y}\left(j\right)=\sum_{i}p\left(i,j\right)$.

Although the statistical features are not visually obvious in the image, however, they proved to enhance the performance of the proposed model. For our application in segmenting obscurant clouds in thermal images, we utilized GLCM to look for lack of texture and structure rather than a repeating pattern or regularities in the image. Therefore, the extracted features were fed into a binary classifier to distinguish the difference between what seems to be a useful patch of the image from what we considered not useful or in other terms obscured. the next section we discuss the choice of parameters for constructing the GLCM, the array of features, the classifier used and the overall model used.

## 4. Methods

### 4.1. GLCM Parameters

#### 4.1.1. Theta θ

Since each pixel has 8 neighboring pixels, we need to choose the directions in which the neighboring pixels are with respect to the reference pixel. We chose the θ to be at 0, and 90 degrees. When θ=0 horizontal pixels in both directions are examined, and when θ=90 the vertical pixels are examined above and below the reference pixel. After testing the performance of the model with all four angles, we did not notice any performance improvements, therefore they were discarded to speed up the feature extraction process.

#### 4.1.2. Block Size σ

The block size or radius is the area that is covered in the GLCM calculations. The radius is fixed at 4.

#### 4.1.3. Distance *d*

The distance is how many pixels away from the reference pixel is the neighbor pixel. It is typically set to match the block size. In other words, the radius is 4 and the pixels being analyzed are at the edges of the local block or two pixels away from the reference pixel P(x, y). Figure 1 demonstrates the pixels considered in the calculation of the GLCM matrix at d=2 when the block size σ=5.

#### 4.1.4. Used Features

After examining the correlation between all GLCM features, almost all features are highly correlated with the variance, so it was excluded from our calculations. and since the ASM is the square of energy we decided to include the energy and exclude ASM. In Figure 2 we demonstrate the correlations of GLCM features. The correlation calculation resulted in excluding the variance, ASM, and dissimilarity. The mean $\mu_{x}$ and $\mu_{y}$ were excluded due to the increase in the dimensionality of the feature vector.

### 4.2. Window Size and Scale Pyramid

The window size in this section is different from the GLCM window size. This window size is to reduce redundant calculations and speed up the processing time for the entire image. The input image is divided into larger windows, which are fed into the classifier to detect any obscurant clouds in extracted windows. Next, the blocks with obscurants are then reanalyzed with a smaller window size. This way we reduce the computational complexity of the model and reduce the processing time.

The initial window size is a square window of the size 128×128, the proceeding sizes are 64×64, 32×32, and 16×16.

### 4.3. SVM

For classification, we used a binary Support Vector Machine (SVM) with a linear kernel. The reasoning behind choosing SVM is that this is a statistical problem of separating two hyper-planes since we are only concerned with binary classification. The aim of SVM is to modify a separating hyper-plane utilizing the separability information of the marginal samples. Given a number of samples that belong to two classes, training a linear SVM means searching for the optimal hyperplane between the two classes and results in the maximum distance between the two classes and the hyperplane, given that the set of samples is linearly separable [31]. After extracting features from both clear and occluded tiles, the data is linearly separable, which made the SVM a good candidate for the classifier due to the simplicity and applicability on linearly separable data.

### 4.4. Data Description

The data used was collected using a thermal camera collecting images of dust clouds. The image then is divided into blocks of size 128×128. The tiles are split into two groups, containing obscurant and clear. Blocks that are partially occluded were not included in either of the two classes. The tiles are then reduced from 16 to bit to 4-bit. The GLCM is then calculated for w×w sub-blocks. Then the 6×1 feature vector is fed into the SVM classifier. The predictions are done on overlapping tiles of the sizes mentioned in the previous section. The overlap in the blocks will produce multi-classification for the overlapping region where the final classification is decided by the higher prediction probability. In the training process we used 1500 tiles for each class, with two scales resulting in approximately 72,000 samples in each class. A few sample images of our dataset along with the ground truth are shown in Figure 3. The complete dataset includes approximately 4000 frames containing occlusions.

### 4.5. Process

In the data collection step, the thermal images were divided into non-overlapping tiles of the size 128×128, only the tiles that were totally occluded and the clear tiles were considered for the future processing and each tile in either group is assigned a label. The next step is to divide the tiles into smaller tiles then the GLCM matrix is constructed for each tile. Then the GLCM feature vector is calculated then passed, along with the label, to the classifier. For the segmentation, a sliding window of size w×w is passed along the image, the GLCM features are calculated, and the window is categorized. Only the windows which are classified to contain an occlusion are then processed again by a sliding window of size $\frac{w}{2}\times \frac{w}{2}$. In the testing phase, we found that the GLCM matrix becomes less representative of the textures in the local tile at tile size of 16×16. Figure 4 represents the overall steps followed in the method implemented.

## 5. Experimental Testing and Evaluation

In the previous section, we covered the type of data, the framework of this model, and the parameters selected to produce the best results. Now we evaluate the segmentation results for the model for varying the window size, distance, and comparing the results with results generated from Gabor texture analysis and Markov Random Field segmentation models. We note that this model is only concerned with binary classification of occluded regions in Long Wave Infrared images.

### 5.1. Evaluation

The empirical evaluation is based on the lapping of classification between the model’s prediction and a user-provided label generated for the same window size. The Ground truths were generated by passing a sliding window over the image and the user decides whether the current window is occluded. We demonstrate the objectiveness in the form of a confusion matrix by counting the true positives *(TP),* true negatives (*TN*), false negatives (*FN*), and false positives (*FP*). We determine the precision, recall, and the *F-score* for the model as demonstrated in Equations (9)–(11):(9)$$precision=\frac{TP}{TP+FP}$$
(10)$$recall=\frac{TP}{TP+FN}$$
(11)$$F-score=2\times \frac{precision\times recall}{precision+recall}$$

In order to determine the efficacy of the GLCM features for different window sizes and the smallest window size that can be used to during the model training stage, we calculate the F-scores for each window size and compare the results. The classification performance for each window size seems to improve as the window size is reduced until the performance drops significantly for window size less than 16×16. The test sizes were chosen to be at increments of 16. Figure 5 demonstrates the F-scores for different window sizes. Finally, in the results Section 5.2, we compare the performance of the proposed model with the performance of the texture analysis methods, Gabor and MRF, discussed in Section 2.2. This comparison stems from the similarity of the process in the proposed approach and other texture analysis, since Gabor features are often applied locally and then passed to a clustering algorithm. In MRF segmentation, the pixels are clustered based on the interrelations between values in the Markovian mesh. We also compare the performance of our approach with the state of the art model (CloudSegNet) introduced in [21], trained on the generated labels in this experiment.

### 5.2. Results

We preform comprehensive testing to our proposed approach on the test set. The testing set is used to calculate the evaluation parameters as shown in Equations (9)–(11), and generate a confusion matrix.

In Figure 6, we demonstrate the quantitative results in the format of a confusion matrix of our classifier model. The model demonstrates the capability to classify local tiles in the images based on only GLCM texture features. We see that the model has a TP rate of 98.7%. On the other hand, it has a false classification rate of 1.3%. These results prove the efficacy of the proposed model at segmenting clouds in LWIR images. However, the model shows a higher rate of false classification of clear tiles. We attribute the higher FN rate to the level of occlusion in the tiles near the edge of the cloud. The previous evaluation was done with fixed parameters with a window size of 32×32 and non-overlapping tiles, the radius and the distance were both set to 5. Next we calculate the effect of the number of feature vectors for a single tile by varying distances and angles when training the classifier in Figure 7. We gathered multiple feature vectors for each local tile at distances of multiplier of 2 (2, 4, 6,..., tilesize). In this scenario, we set tile size to 32×32. Considering the results displayed in Figure 7, we noticed that including features of the same tile at different distances does not enhance the performance of our model. Therefore, we set the final model to have a fixed distance and radius of 5 due to the slight improvement in the performance at distance d=4. This step was necessary to reduce the complexity of the model and significantly reduce training time for the final model.

In Figure 8, we display some of the resultant frames from the testing stage. In the upper row, are the input images, the middle row are the results from using a larger window size (64×64) which results in the tiling effect in the predicted mask. Finally in the bottom row, the results from using window size of 32×32.

Next, we compare the results from our proposed approach with popular texture segmentation approaches, mainly Gabor texture segmentation, and Markov random field. We choose the most common and highly suggested methods due to their demonstrated effectiveness in segmenting textures. In addition, we use the labels to train an FCN to perform the binary segmentation on our images. We show the results from these trials in Figure 9. We notice that the corresponding mask generated by the Gabor segmentation model corresponds to the K-Means clustering error caused by the clustering based on the pixel intensities in the filtered image bank generated by the Gabor filters. Similarly, the MRF segmentation result shows that it occlusion classification can mistake several regions of similar pixel intensities as an occluded region. Our approach, on the other hand, is capable of distinguishing areas where the dust cloud becomes less occluding and the scene behind it is visible as shown in Figure 9. The quantitative results from this evaluation is presented in Table 1. In the results table, we evaluated each approach at various occlusion levels. The proposed approach demonstrates consistent robustness at the three levels of occlusions, while we notice discrepancies in the performances in other approaches. Since the proposed model was is a classifier trained to distinguish the texture fabric of the occlusion in LWIR images, it will perform better than other models (i.e., clustering Gabor responses), since all other approaches rely on either the difference or the similarity in adjacent regions in the image to group them in a single category. As demonstrated in the table below, the proposed approach out performs the state of the art and the popular methods qualitatively, quantitatively, and in terms of the training data required (as the case for all Deep Learning models).

## 6. Conclusions

We presented a framework for segmenting obscuring clouds in Long Wave Infrared images as a part of a project to provide a full semantic segmentation algorithm. The results are the completely applicable to autonomous vehicles as many of the major challenges facing widespread autonomous vehicle adoption are addressed with improved segmentation processes. Specifically, the ability classify and segment cloud-like obscurrants will enable enhanced autonomous vehicle performance in compromised operating environments such as fog, haze, or smoke. As such, the motivation behind the work can be encapsulated by the lack of labeled data and the accurate deep learning architectures to perform the task at hand. When we attempted CNN based models for segmenting clouds such as CloudSegNet, the model performed poorly since it was designed and tuned to deal with visible color images, and the nature of the data the model was trained with has only sky as the background. The mentioned obstacles inspired the search for a more accurate and easier approach to classify obscurants pixels in the frame. The proposed method is capable of classifying tiles of varying sizes into two categories, containing occlusion or clear. We demonstrated in the results Section 5.2 that our approach can texture-segment the frame far more accurately than famous texture analysis techniques. We were able to achieve a precision of 93% in 16 s per frame. Where Gabor segmentation took 140 s to perform the same task with sub par accuracy. Additionally, since we are dealing with thermal images, this method, unlike other methods, can perform as accurately regardless of time of day, where other methods require further pre-processing of the images. This research is one of the paving steps towards building a pan-optic segmentation system which is capable to perform in severe weather conditions and regardless of the illumination levels. The work presented in this paper is important to enhance the segmentation for other categories, which in its turn enhance the understanding of the scene semantics while driving in severe weather conditions. the In future work, we plan on expanding the model to provide a multi-label semantic segmentation, that can run in real time.

## Figures and Tables

**Figure 1 jimaging-08-00266-f001:**
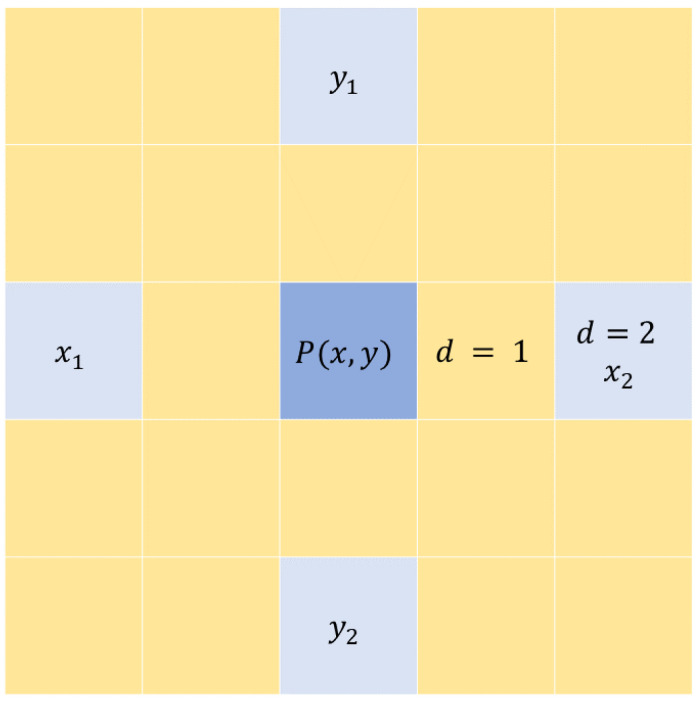
The GLCM matrix is calculated for center pixel P(x,y) using the pixels at d=2.

**Figure 2 jimaging-08-00266-f002:**
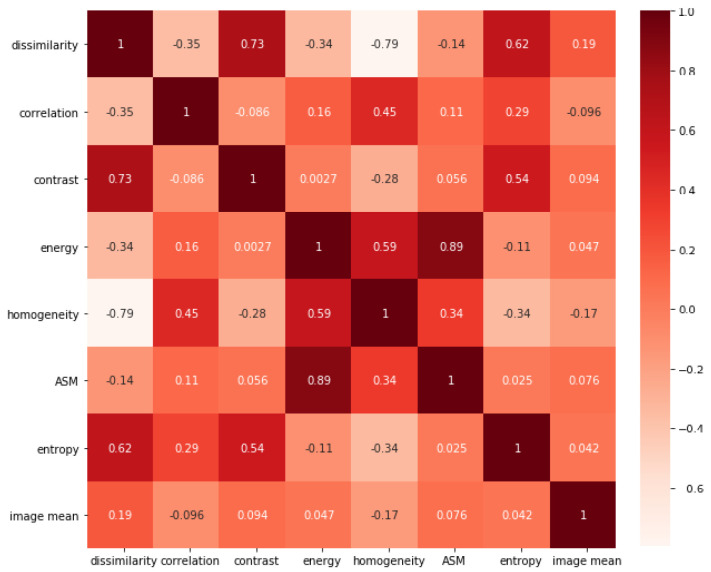
Correlations in the initial GLCM feature vector.

**Figure 3 jimaging-08-00266-f003:**
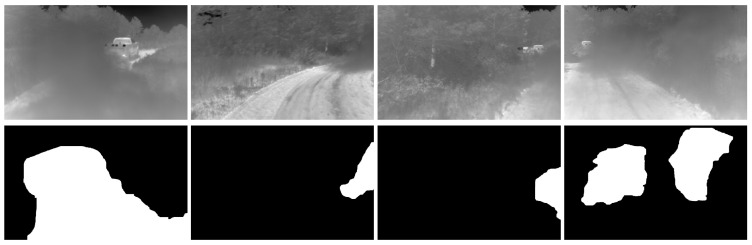
Sample images from our dataset (**top row**) along with corresponding occlusion segmentation ground truth (**bottom row**).

**Figure 4 jimaging-08-00266-f004:**
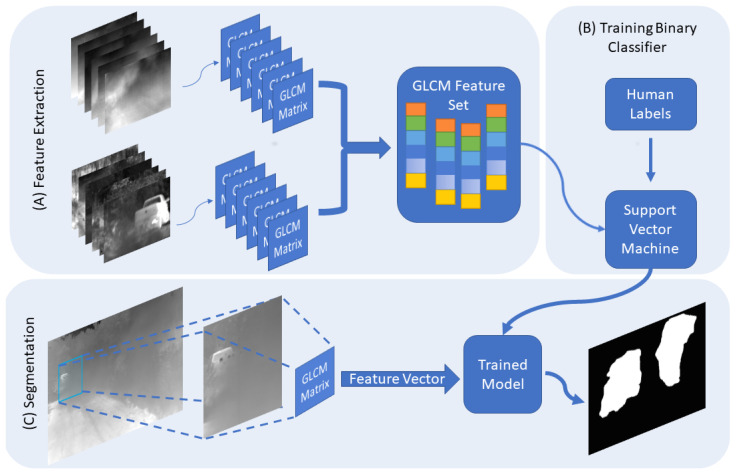
In (**A**), the features are extracted by picking 128×128 tiles which are either completely occluded, or clear. Then we extract the GLCM features from tiles with varying sizes. Next, we train the classifier using the feature vectors and the labels from the tiles collected (**B**). To segment the obscuring cloud (**C**), a larger sliding window passes along the image and generates the GLCM features for the window then classifies it. Then a smaller window will only process the tiles that include obscurants. The scaled sliding window process will continue until the stopping size is reached.

**Figure 5 jimaging-08-00266-f005:**
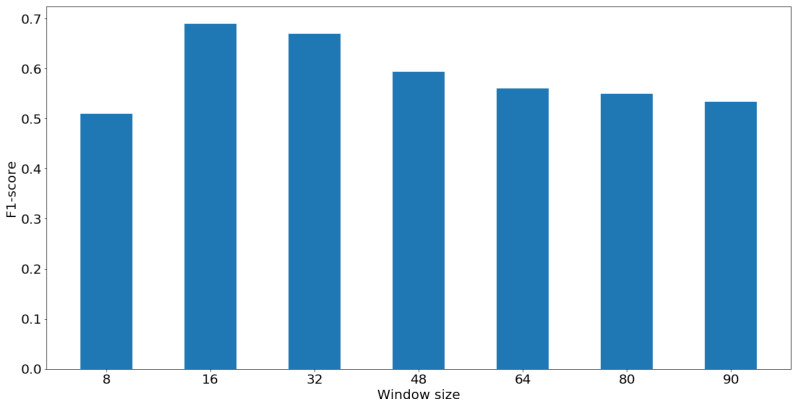
Classification performance evaluation for different radii to generate the GLCM matrix.

**Figure 6 jimaging-08-00266-f006:**
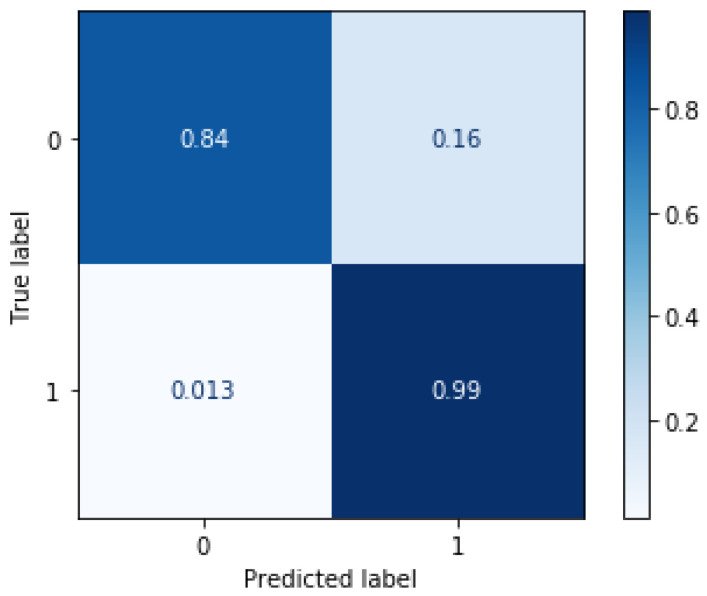
Classification performance evaluation. The normalized confusion matrix demonstrates the performance of the proposed model in classifying tiles containing obscurant clouds. The model tends to miss-classify tiles with tiles near the edge of the obscurant clouds since the human labels considered the edges clear although containing very light occlusion.

**Figure 7 jimaging-08-00266-f007:**
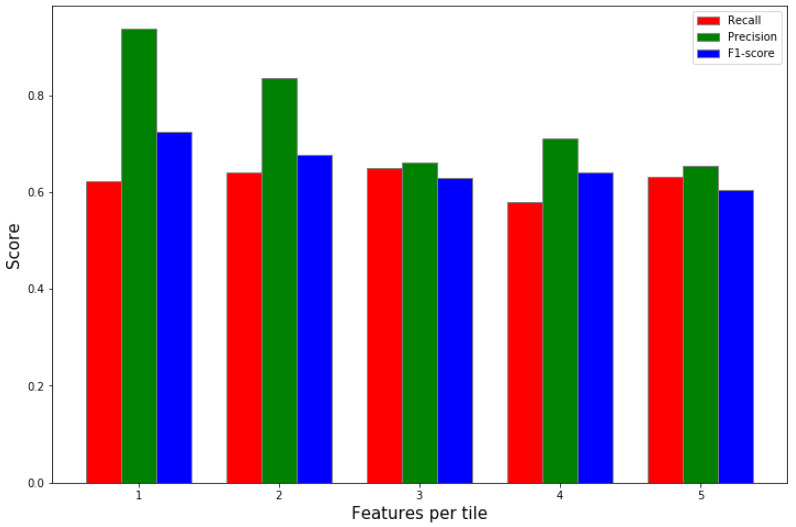
Comparing performance when including several feature vectors per tile calculated at different distances from reference pixel. The negligible improvement in performance demonstrates the irrelevance in the calculated features at distances larger than 3 tiles. However, we notice an increase in the performance, although trivial, when including two vectors (at d=2, and d=4).

**Figure 8 jimaging-08-00266-f008:**
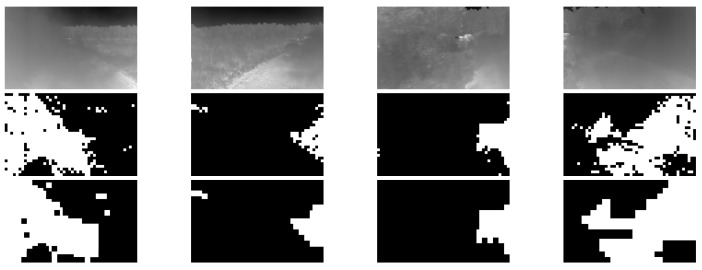
Qualitative results from running tests with varying the window size. The top row is the input image, the middle row shows the results with window size of 64×64, and the bottom row shows the masks generated from 32×32 tiles.

**Figure 9 jimaging-08-00266-f009:**
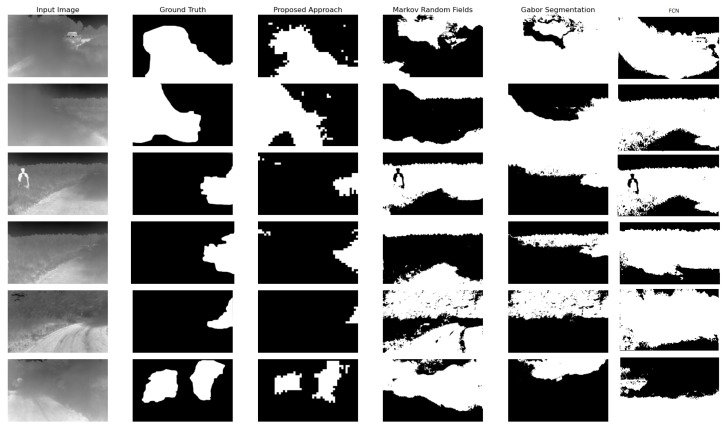
Sample results with the corresponding ground truth comparing the results from the proposed approach, with results from Gabor and MRF segmentation. As displayed in the results above, the proposed approach provides accurate results compared to the other two methods for texture analysis. Although texture analysis is very accurate in RGB and grayscale images, it can produce far less accurate results when tested in thermal images.

**Table 1 jimaging-08-00266-t001:** Table of the results using binary classification on our dataset. The evaluation was done over three classes of occlusion densities: 0, 30 and 90 %. The metrics are calculated for all three classes Precision, Recall, F-score, and Miss-classification represented as M.

Occlusion Level	0 %				30%				90%				Run Time
**Methods**	**Precision**	**Recall**	**F-Score**	**M**	**Precision**	**Recall**	**F-Score**	**M**	**Precision**	**Recall**	**F-Score**	**M**	
Our approach	0.88	0.59	0.70	0.21	0.93	0.62	0.72	0.2	0.9	0.79	0.80	0.07	8 s
Gabor	0.10	0.31	0.11	0.70	0.16	0.40	0.20	0.69	0.60	0.56	0.73	0.30	138 s
MRF	0.28	0.46	0.34	0.72	0.29	0.47	0.34	0.71	0.35	0.49	0.50	0.69	45
FCN	0.65	0.50	0.59	0.30	0.61	0.44	0.51	0.62	0.62	0.49	0.47	0.55	2

## Data Availability

https://github.com/mbhssein/GLCM_cloud_segmentation (accessed on 19 May 2022).

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
