# Peer review of "Obscurant Segmentation in Long Wave Infrared Images Using GLCM Textures"

_2313-433X, 2022, doi:10.3390/jimaging8100266_

Round 1

Reviewer 1 Report

The paper present an interesting application for the classification of infrared onboard cameras. Even if the methodology is correctly applied as the research, the authors have chosen to compare very different class of segmentation/classification methods.

Line 130, the Agency is DARPA ".. challenge of DAPRA is also studied.."

Line 285, "Supporting Vector Machine" is an awkward denomination for the SVM subchapter. Also, MRF is usualy use as a Markov Random Field, not Fields.

As for the comparison, I think that comparing a Machine Learning with a MRF or a Gabor wavelet is like comparing a Mercedes, a Ford and a Tata... There are more up to date algorithms and methods that can support the test of the proposed method.  Your are extracting a lot of features from the image, Gabor function is using only two. Disproportionate comparison.

Author Response

  1. The paper present an interesting application for the classification of infrared onboard cameras. Even if the methodology is correctly applied as the research, the authors have chosen to compare very different class of segmentation/classification methods: 
    We chose to compare our approach which relies on texture features (GLCM) and a classifier with similar texture segmentation approaches which are built in a similar manner, for example Gabor textures are done by applying several filters on the image then k-means is used to cluster similar responses. MRF clusters pixels based on the probability distribution of the grayscale image. 
    However, we added an deep learning segmentation approach since we mention deep learning models in the related work section. 
  2. Line 130, the Agency is DARPA ".. challenge of DAPRA is also studied.."
    Fixed.
  3. Line 285, "Supporting Vector Machine" is an awkward denomination for the SVM subchapter. Also, MRF is usually use as a Markov Random Field, not Fields. 
    Also fixed
  4. As for the comparison, I think that comparing a Machine Learning with a MRF or a Gabor wavelet is like comparing a Mercedes, a Ford and a Tata... There are more up to date algorithms and methods that can support the test of the proposed method.  Your are extracting a lot of features from the image, Gabor function is using only two. Disproportionate comparison.
    First item on this list explains our reasoning behind this comparison.

Reviewer 2 Report

The authors proposed an approach for segmenting obscuring clouds in long wave infrared images. Although the obtained results are encouraging, the paper has the drawbacks described below that need to be solved before reconsidering it.

 1. Section 2 (“Related Works”) needs to be revised to include more recent works (<5 years). In addition, the text of this section needs to be concise, making it clear in which the proposed approach differs from those found in the literature and what are the contributions of the work. Finally, the question presented at the end of this section ("What property do we look for to segment such a random and, inconsistent object when segmenting obscurants?") is out of context and has not been answered in the results or conclusions sections.

2. The following points in section 3 need to be reviewed: i) make the difference between radius and distance clear, since in Haralick's paper only distance is considered; ii) is the Angular Second Moment equation correct?

3. In section 4.3 the authors say: "Given a number of samples that belong to two classes, training a linear SVM means searching for the optimal hyperplane between the two classes and results in the maximum distance between the two classes and the hyperplane, given that the set of samples is linearly separable [32]". Can the authors state that the data used in the experiments are linearly separable?

4. The results presented in section 5 are insufficient to demonstrate the robustness of the proposed approach. In addition, they need to be better presented and discussed. Some other problems in this section: i) is the equation 11 correct?; ii) In the page 11 the authors say “Next we calculate the effect of calculating several feature vectors at varying distances and angles when training the classifier in Figure 6.”, but Figure 6 does not demonstrate this; iii) The authors mention accuracy of the proposed model, but according to section 5.1 this measure was not considered; iv) The results presented in Table 1 was not discussed; v) In the page 12 the authors say “The slight change in performance demonstrates the irrelevance in the calculated features at distances larger than 4 tiles”, but this is not clear.

5. All acronyms used must have their meanings described when it is appearing the first time in the text.

6. Figures and Tables must be cited in the text before appearing.

7. The way of citing works from the literature needs to be reviewed and standardized. 

Author Response

  1.   Section 2 (“Related Works”) needs to be revised to include more recent works (<5 years). In addition, the text of this section needs to be concise, making it clear in which the proposed approach differs from those found in the literature and what are the contributions of the work. Finally, the question presented at the end of this section ("What property do we look for to segment such a random and, inconsistent object when segmenting obscurants?") is out of context and has not been answered in the results or conclusions sections.
    • In the first draft we reviewed recent algorithms (2019) called CloudSeg. We added a recent benchmarking paper (2020) that evaluated all the cloud segmentation algorithms we investigated.
    • Removed the question at the end of the related work section since it was deemed unnecessary by the following section.
  2. The following points in section 3 need to be reviewed: i) make the difference between radius and distance clear, since in Haralick's paper only distance is considered; ii) is the Angular Second Moment equation correct?
    • The mention of distance was removed since it was not mentioned in the Haralick's paper.

  3. In section 4.3 the authors say: "Given a number of samples that belong to two classes, training a linear SVM means searching for the optimal hyperplane between the two classes and results in the maximum distance between the two classes and the hyperplane, given that the set of samples is linearly separable [32]". Can the authors state that the data used in the experiments are linearly separable?
    • Addressed
  4. The results presented in section 5 are insufficient to demonstrate the robustness of the proposed approach. In addition, they need to be better presented and discussed. Some other problems in this section: i) is the equation 11 correct?; ii) In the page 11 the authors say “Next we calculate the effect of calculating several feature vectors at varying distances and angles when training the classifier in Figure 6.”, but Figure 6 does not demonstrate this; iii) The authors mention accuracy of the proposed model, but according to section 5.1 this measure was not considered; iv) The results presented in Table 1 was not discussed; v) In the page 12 the authors say “The slight change in performance demonstrates the irrelevance in the calculated features at distances larger than 4 tiles”, but this is not clear.
    • Equation 11 corrected
    • Figure 6 demonstrates the performance when including different number of feature vectors. language was corrected.
    • "performance" was the word meant in that context, changed the phrasing.
    • V) addressed
  5. All acronyms used must have their meanings described when it is appearing the first time in the text. Addressed
  6. Figures and Tables must be cited in the text before appearing.
    • Latex formatting can be slightly challenging. However, rearranged the text as much as possible.
  7. The way of citing works from the literature needs to be reviewed and standardized.  This paper was submitted for language and organizational review and changes were made.

Round 2

Reviewer 1 Report

The authors have replied to the comments and solved some of the issues. Overall, the paper is good to go. Anyhow, in the future, it is recommended to compare with state of the art methods and referenced ones, using widely used benchmarks.

Author Response

We have reviewed the most recent, state of the art approaches that fall under the scope of cloud segmentation in the references 17-24. Other recent segmentation approaches are out of scope since they do not perform at all on our type of data.

Reviewer 2 Report

I have reviewed the revised version of the manuscript and I noticed that the authors only partially answered my requests. The following points are still unresolved:

1. The literature review presented in section 2 has improved somewhat, but remains poor.

2. My question about the radius and distance parameters has not yet been fully resolved, as they are still mentioned in subsections 4.1.2 and 4.1.3 and on page 10 (lines 304 to 306).

3. Is the Angular Second Moment equation correct? (this question was not answered by the authors)

4. On Figure 6, shouldn't the abscissa axis express the parameter "distance"?

5. There are still acronyms in the text without a description of their meanings. Examples: DARPA and LWIR. In addition, there are still problems with citing references. Examples: D. Tulpan and Dev (section 2.1 - lines 88 and 96).

6. Finally, the authors did not respond to my questioning about the results being insufficient.

Author Response

  1. The literature review presented in section 2 has improved somewhat, but remains poor.

In the literature review in section 2, we removed irrelevant topics (topics within the general scope of the project but not necessarily the scope of the paper) and kept the relevant topics, and expanded the segmentation to current Deep Learning approaches, and added them to the evaluation section

  1. My question about the radius and distance parameters has not yet been fully resolved, as they are still mentioned in subsections 4.1.2 and 4.1.3 and on page 10 (lines 304 to 306).

The difference is that the block size is the size of the local area being analyzed (window size for creating the glcm matrix) and the distance (d) is the distance from the center pixel in the block we considered as the “neighboring” pixel. For example, we were working with a 5X5 block, the center pixel is 2 pixels away from the edge of the block and that is what we considered as our distance d. since it is computationally expensive to run this analysis for every pixel , skipping pixels for higher resolution images helps improving the speed of the approach. The numbers were incorrect in the first and second versions but are correct in this version. Added figure 1 to visually demonstrate

  1. Is the Angular Second Moment equation correct? (this question was not answered by the authors).

This mistake was corrected based on the GLCM calculations in the reference paper.

  1. On Figure 6, shouldn't the abscissa axis express the parameter "distance"?

In figure 5 (previously figure 6) we run the analysis by varying the window size (W) for the classifier. So we split the image into smaller WXW patches. We treat every smaller window as an image, run analysis, and then classifier. We evaluated the performance on single patches to find the optimal window size. This window size is the parameter mentioned in section 4.2.   

  1. There are still acronyms in the text without a description of their meanings. Examples: DARPA and LWIR. In addition, there are still problems with citing references. Examples: D. Tulpan and Dev (section 2.1 - lines 88 and 96).

The DARPA text was a part of the irrelevant text so it was removed and fixed the LWIR acronym.

  1. Finally, the authors did not respond to my questioning about the results being insufficient.

Added evaluations for different levels of occlusions and included deep learning model in the evaluation.